# Pediatric Lower Urinary Tract Dysfunction: A Comprehensive Exploration of Clinical Implications and Diagnostic Strategies

**DOI:** 10.3390/biomedicines12050945

**Published:** 2024-04-24

**Authors:** Matjaž Kopač

**Affiliations:** Department of Nephrology, Division of Pediatrics, University Medical Centre Ljubljana, Bohoričeva 20, 1000 Ljubljana, Slovenia; matjaz.kopac@kclj.si; Tel.: +386-15229626

**Keywords:** voiding disorders, urinary incontinence, lower urinary tract, children, voiding diary, urodynamics

## Abstract

Lower urinary tract dysfunction is clinically important because it may cause urinary tract infections, mainly due to accumulation of residual urine, and adversely affect renal function. In addition, it may cause urinary incontinence, strongly affecting the child’s quality of life. The function of the lower urinary tract is closely associated with function of the bowel because constipation is commonly present with bladder dysfunction. The interplay between the lower urinary tract and bowel function, coupled with common conditions such as detrusor overactivity and voiding dysfunction, requires a nuanced diagnostic approach. Detrusor overactivity, a benign but socially harmful condition, is the principal cause of daytime urinary incontinence in childhood. It needs to be differentiated from more serious conditions such as neurogenic bladder dysfunction or urethral obstruction. Voiding dysfunction, a habitual sphincter contraction during voiding, is common in children with detrusor overactivity and may be self limiting but may also result in residual urine and urinary tract infections. It may resemble, in severe cases, neurogenic bladder dysfunction, most often caused by spinal dysraphism, which very often leads to recurrent urinary tract infections and high intravesical pressures, jeopardizing renal function. A voiding diary is crucial in the initial evaluation of lower urinary tract function in children.

## 1. Introduction

Pediatric lower urinary tract dysfunction (LUTD) encompasses a spectrum of conditions that significantly affect a child’s health and well-being. This article provides an in-depth exploration of the clinical relevance of LUTD, focusing on its association with urinary tract infections (UTIs), renal function, and the profound impact on a child’s quality of life [1,2,3]. The intricate relationship between the lower urinary tract and bowel function is emphasized, particularly in the context of constipation. Detrusor overactivity (DO), a common yet socially harmful condition, is identified as a leading cause of daytime urinary incontinence in childhood, necessitating careful differentiation from more severe conditions. Additionally, the article delves into voiding dysfunction, its manifestations, and potential complications, highlighting the importance of a comprehensive diagnostic approach, including voiding diaries, urodynamics, and various imaging modalities [4,5,6,7]. The purpose of the LUTD evaluation in children is to differentiate whether there is an abnormality in filling or emptying of the bladder or both and whether there is a bladder function abnormality; in the latter case, the evaluation should determine the underlying cause. In addition, the assessment should distinguish between organic (anatomical or neurogenic) and functional causes of LUTD since therapy differs depending upon the cause. The evaluation for most children with LUTD can be performed with a thorough history, physical examination, and noninvasive testing, such as a urinalysis and urine culture. However, in selected children, more extensive investigations include imaging studies and measurement of urinary flow and post-void residual. In most complex LUTD pediatric patients, urodynamic studies may be required in order to diagnose the underlying etiology and determine the most appropriate therapy [8].

The aim of this article is to provide a comprehensive overview of pediatric LUTD, including its clinical relevance, available diagnostic and therapeutic tools, differentiation of specific LUTD subtypes, implications for prognosis as well as to offer clinically useful insights into the complex interplay between lower urinary tract and bowel dysfunction and various clinical outcomes. Bearing this in mind, it should help physicians and other healthcare professionals, involved in the management of children with LUTD, to implement these findings in clinical practice in order to improve prognosis of these patients.

The literature search strategy included searching for terms in different medical databases, such as PubMed, UpToDate (available in our hospital), and the author’s personal archive of peer-reviewed articles and textbooks, published in English up to 31 December 2023, using the search terms “voiding disorders, urinary incontinence, lower urinary tract, children, voiding diary, urodynamics and urinary tract infections”.

## 2. Clinical Relevance of Pediatric Lower Urinary Tract Dysfunction

### 2.1. Urinary Tract Infections (UTI)

Pediatric LUTD is clinically relevant due to its association with recurrent UTIs. The accumulation of residual urine, a common consequence of LUTD, creates a conducive environment for bacterial growth, increasing the risk of UTIs. Recurrent infections not only pose immediate health risks but also elevate the likelihood of long-term complications, including renal damage [1]. In addition, UTIs with concurrent vesicoureteral reflux (VUR) are often seen in children with LUTD; in the latter, the therapy of VUR is more challenging [8] with a higher failure rate of surgical correction, increased incidence of UTI, and longer time for VUR resolution [9]. Treatment of LUTD, particularly overactive bladder, has been shown to improve spontaneous resolution of VUR, indicating that overactive bladder has a role in the development of VUR [9].

For this reason, urine analysis and urine culture should be performed in children presenting with LUTD in order to screen for UTI [9]. A causal relationship between UTI and LUTD has not been confirmed but it is accepted that UTIs may lead to LUTD [10] as well as LUTD predisposes children to recurrent UTI and kidney injury [11]. In addition, the risk of UTI is increased in children with incomplete bladder emptying due to dysfunctional voiding or underactive bladder as well as in children with primary neck bladder dysfunction with opening of the bladder neck during contraction [9].

### 2.2. Renal Function Impairment

Adverse effects of LUTD extend to renal function, emphasizing the significance of timely diagnosis and intervention. Accumulation of residual urine may lead to increased intravesical pressure, jeopardizing kidney function and potentially causing irreversible damage. Understanding the impact of LUTD on renal health is crucial for developing effective management strategies [2]. In addition, chronic kidney disease has been reported in children with LUTD, recurrent UTI and VUR [9].

### 2.3. Association with Bowel Function

The lower urinary tract’s function is intricately associated with that of the bowel, with constipation commonly coexisting with bladder dysfunction in pediatric populations. Understanding this association is paramount in unraveling the complexities of LUTD and devising targeted interventions. For this reason, information regarding the child’s bowel habits should be obtained. This includes the frequency of bowel movements, the consistency, size, and caliber of stool, presence of painful defection and any history of stool withholding behavior, and fecal incontinence or soiling [8,12]. There are several potential mechanisms that may explain the above-mentioned association: the bowel and the lower urinary tract share a large part of their neuromuscular innervation, fecal impaction causes the rectum to be continuously dilated and thus compresses the bladder from posterior aspect, and, in addition, behavioral issues (e.g., voiding postponement) may not only influence the bladder but also bowel habits. Therefore, constipation may be a main contributing factor in urinary stasis with recurrent UTIs, DO, urinary retention, and voiding dysfunction. Moreover, most children with neurogenic bladder also suffer from neurogenic bowel disturbance. Treatment of associated constipation can significantly improve the course of patients with overactive bladder. For this reason, these children need to learn proper defecation patterns, which leads to shrinkage of the distended rectum, enabling it to function as a signal space, not a storage organ. This usually takes patience and time. Using daily mini-enemas at start can be very useful (if tolerated), followed by polyethylene glycol daily, in order for children to achieve soft bowel movements without encopresis. In addition, proper hydration, physical activity, healthy diet, and regular bowel movements are strongly recommended [5].

### 2.4. Detrusor Overactivity and Voiding Dysfunction: Clinical Manifestations and Complications

Detrusor overactivity (DO), a benign yet socially harmful condition, is the leading cause of daytime urinary incontinence in childhood. Distinguishing DO from more severe conditions such as neurogenic bladder dysfunction or urethral obstruction is crucial for appropriate management. Children with DO have involuntary detrusor contractions during the filling phase, with increased bladder pressure exceeding 15 cm of water. However, there are no other voiding abnormalities, and these children have normal sphincter activity during both the filling and voiding phases and normal detrusor contractions during voiding [8]. Studies in children with voiding dysfunction symptoms revealed DO, detected by urodynamic investigation, in 52–58% of patients compared with only 5–18% of asymptomatic children [11,13].

Voiding dysfunction, characterized by habitual sphincter contraction during voiding, is common in children with DO. While it may be self-limiting, severe cases may lead to complications such as residual urine and UTIs. In extreme instances, the manifestation of voiding dysfunction closely resembles neurogenic bladder dysfunction, often associated with spinal dysraphism, posing a significant threat to kidney function. However, dysfunctional voiding, a similar yet different term, occurs in children with a neurologic lesion, often called detrusor sphincter dyssynergy, or children without a neurologic lesion, named non-neurogenic dysfunctional voiding. There is abnormal contraction of the urethral sphincter during voiding but normal detrusor function during both the filling and voiding phases and normal sphincter contraction during the filling phase in all of them [8].

## 3. Diagnostic Approaches: A Comprehensive Toolbox for Evaluation

### 3.1. History

A detailed history is of utmost importance in diagnostic evaluation of LUTD. The history should be adapted to the age and the appropriate stage in the development of bladder control of the patient. The history should focus on the following [8]:voiding schedule, including information about the frequency of voids and the frequency of incontinent episodes (in toilet-trained children). An estimation of the voided volume should be obtained as well, if feasible. Large-capacity bladders are present in children with underactive bladder or polyuria, presenting with large volumes of voided urine [8];symptoms of bladder dysfunction, such as urgency, painful urination, hesitancy, holding maneuvers, dribbling, straining, and an intermittent or weak urinary stream. Table 1 presents the definitions for these symptoms [2,3,9];there are several surveys and questionnaires available in order to assess daytime incontinence, bowel habits, urgency, voiding habits, dysuria, and quality of life. They have been shown to be equivalent when evaluating response to treatment and to correlate with clinical impression of physicians; however, the mean symptom scores from these surveys and questionnaires were higher than the physician’s rating for symptom severity;family history that should screen for any kidney or urologic disorders including LUTD and also the age that other family members achieved urinary continence in order to confirm or exclude a familial maturational delay in achieving bladder control;perinatal and neonatal history, searching for evidence of any perinatal or neonatal insult, such as perinatal anoxia or congenital infection that could impact the central and peripheral nervous system normal coordination of bladder function;diet intake, including information about the amount and type of fluid intake. Excessive fluid intake and/or fluid intake during the nighttime, for example, may suggest diabetes mellitus, polyuria due to a concentrating defect, or, rarely, primary polydipsia;neurodevelopment delay and psychological disorders may delay gaining voluntary bladder control. There is an increased risk of LUTD in children with psychological disorders, such as attention deficit hyperactivity disorder (ADHD), depression, and anxiety. However, the majority of children with LUTD do not have behavior problems;functional causes of LUTD often originate from behavioral issues arising from toilet training or personal stress, arising from a conflict between the parent or caregiver and child;toilet training history, especially if it was prolonged, delayed, stressful, or with a period of dryness after toilet training. Anatomic causes of urine incontinence, such as an ectopic ureter, typically lack a period of complete dryness after toilet training [8].

### 3.2. Voiding Diary

The use of a voiding diary is emphasized as a crucial tool in the initial evaluation of lower urinary tract function in children, in addition to history and physical examination. This detailed record of voiding patterns and associated symptoms provides valuable insights, aiding in the diagnosis and management of LUTD [4]. A three-day voiding diary is useful for obtaining a record of urinary voiding and defecation patterns. It should include the time and volume of each void, every episode of incontinence, fluid intake, every defecation, and any episode of fecal soiling [8].

Uroflow is especially useful for children old enough to void on command, especially in cases of weak stream or the need to strain to void or with recurrent UTIs secondary to incomplete bladder emptying or therapy-resistant incontinence. For this purpose, a child will void on a toilet that measures the flow rate, and immediately afterward, the residual urine in the bladder is assessed with ultrasound (US) device. A pathological curve needs to be repeated by uroflowmetry studies before conclusions may be drawn. When a noninvasive screening of the LUT function of infants and children before the age of bladder control is indicated, a 4 h voiding observation is the preferred method. For this purpose, the child is allowed to play and eat freely for four hours while every void is documented by weighing diapers, and US is used to measure post-void residual. In the presence of normal bladder function, there should be at least one complete void without residual urine during the observation period [5]. Otherwise, basic principles of investigations and treatment for children with gained bladder control apply for infants as well; however, they must be tailored to their age and stage of development.

### 3.3. Physical Examination

The purpose of physical examination is to detect neurologic, urologic, and other abnormalities. It should include examination of various parts of a body, as presented in Table 2 [8]. It is worth mentioning that any abnormality of the neurologic examination may suggest a neurologic lesion, also affecting the bladder function due to an integrated neural network coordination [8].

In addition, sexual or physical abuse should be considered sometimes during examination since LUTD dysfunction may be a presenting sign for child abuse. These children may present with urgency, urge and stress incontinence, voluntary voiding postponement, intermittent urination associated with straining, recurrent UTIs, and constipation with or without encopresis. They are evaluated sometimes for these symptoms when they do not improve with standard treatment regimens. The possibility of sexual abuse should be considered in the evaluation of a child with new onset of LUTD or of an adult presenting with long-term LUTD with no neurological or obstructive etiology. Sexual abuse occurs in all socioeconomic settings and friends, strangers, or family members may be the perpetrators. The latter are often aware of the abuse but are unable to discern its relationship to other problems, such as LUTD. Therefore, history and physical examination are important in the evaluation of sexual abuse, with special attention on simple clues. The history of sexual abuse is often elicited during the urodynamic evaluation. It is important to ask the parent and the child about possible sexual abuse since some children who were abused by a family member or friend of the family will often discuss the matter more easily in the absence of parents. For children between three and six years of age, the use of line drawings, dolls, or other aids may be helpful. A structured interview for sexual abuse is provided in many hospitals and professional associations. Physical examination is very important as well, including a brief neurological examination in order to rule out a neurological cause. Assessment is best performed on a relaxed child. However, signs of acute damage may resolve within six weeks of the abuse and wounds may heal completely without sequelae. An anorectal examination should be performed to look for tears, fissures, bleeding, and relaxation of the anus in both sexes. All children with abnormal ano-genital examinations should be referred to a pediatrician familiar with the assessment of sexual abuse and parents should be informed that the examination is abnormal and further evaluation is required; however, no accusations should be made at this point. The treatment of LUTD in children with a history of sexual abuse consists of timed voiding, double voiding regimens for those with increased post-void residuals, voiding diaries, behavior modification, positive feedback using uroflowmetry, regular bowel regimen, anticholinergics (when indicated), psychiatric counseling, and, only exceptionally, clean intermittent catheterization. In this way, symptoms improvement or resolution can be expected in most of these children [14].

### 3.4. Laboratory Investigations

Initial laboratory testing includes urinalysis and urine culture. Results of urinalysis, optimally performed on a first morning void, may reveal diseases due to a renal concentrating defect or glycosuria due to diabetes mellitus where polyuria is a main symptom. A urine culture should also be obtained, especially if leukocyte esterase, pyuria, and nitrite are present in the urinalysis, due to increased risk of UTI in children with LUTD. Serum laboratory studies, such as creatinine concentration, are usually not performed initially since kidney function impairment is rare in children with a normal urinalysis. However, abnormal results of urinalysis, such as proteinuria or a low specific gravity, indicates kidney disease, malformation, or injury and demands a serum electrolytes and creatinine concentration measurement in order to estimate the glomerular filtration rate [8].

### 3.5. Imaging Studies

Ultrasound (US) of a kidney and bladder, a noninvasive investigation, is the most commonly used imaging study in the evaluation of children with LUTD and should be done in every child with a suspected neurologic or anatomical lesion, UTI, or symptoms indicative of an obstructive uropathy, such as weak or interrupted urinary stream. US can give various information, including detection of anatomical abnormalities (hydronephrosis, duplicated collecting system with or without an ectopic ureter, kidney scarring, to name just a few of them), measurement of post-void residual volume (suggesting incomplete bladder emptying when a volume exceeds 20 mL after repeat measurement, present in underactive bladder, for example) and measurement of bladder wall thickness, indicating (if thickened) outlet obstruction due to an anatomical or functional abnormality or, most commonly, overactive bladder [8].

Voiding cystourethrogram (VCUG) is a contrast study, using either X-ray or ultrasound contrast agent, that involves urethral catheterization and is able to assess the bladder during the filling and voiding phases to detect VUR and posterior urethral valves (PUV) and to give information on bladder shape, capacity, and bladder emptying. We use it in children with UTI and in boys suspected of having PUV [8]. Contrast-enhanced ultrasonography (US) has become an important supplementary tool in many clinical applications in children. Contrast-enhanced voiding urosonography has proved useful in routine clinical practice in children because it is practical and radiation-free. Intracavitary contrast-enhanced US is a real-time imaging modality similar to fluoroscopy with iodinated contrast agent. The US contrast agent solution is administered into physiological or non-physiological body cavities, where it can be used for many clinical applications. It offers excellent real-time spatial resolution and allows for a more accurate delineation of various details of anatomical relations [15].

Videourodynamic (VUD) investigation combines VCUG and urodynamics into one study, which allows simultaneous visualization of the urinary tract morphology and presence of VUR by ionizing radiation together with the measurement of sensation, bladder capacity, compliance, and detrusor pressure during bladder filling and voiding using one double lumen catheter. Fluoroscopic videourodynamics has concerns regarding radiation exposure. This can be solved by replacing fluoroscopic VCUG with contrast-enhanced voiding urosonography (ceVUS) using second-generation contrast media and harmonic imaging, which is a radiation-free and highly sensitive imaging modality that can better detect VUR, including intrarenal reflux combined with urodynamic disorders associated with VUR [16].

Magnetic resonance imaging (MRI) should be performed in children with neurologic signs and symptoms in order to look for occult neurologic lesions. It is worth mentioning that a normal physical exam does not necessarily exclude occult spinal cord disorders. This has been proven in a study in children with LUTD, refractory to medical therapy and a normal physical examination, where 39% of them had pathologic findings on MRI [17]. According to that, a lumbosacral MRI is indicated in children suspected of having a neurologic abnormality and in those who do not respond to therapy or manifest urodynamic findings consistent with a neurologic defect [8].

Altogether, these techniques offer valuable insights into the anatomical aspects of the lower urinary tract. This is especially important in cases resembling neurogenic bladder dysfunction, particularly those associated with spinal dysraphism, where a thorough evaluation is necessary [7,8,13].

### 3.6. Urodynamics

Urodynamic studies play a vital role in differentiating between various lower urinary tract dysfunctions, contributing to a more nuanced understanding of the condition. These studies involve invasive (cystometry) and non-invasive measurements (uroflowmetry, residual urine measurement, and voiding observation) to assess bladder and sphincter function [6].

Uroflowmetry (measurement of urinary flow) can give information about urine flow pattern that is often diagnostic of an etiology, thus enabling us to avoid performing more invasive urodynamic testing. In addition, it gives information about the emptying phase of the bladder; however, it can give no data about the filling phase. For this purpose, children are asked to wait until they feel a strong desire to void and then to void into a special vessel, producing a urinary flow curve, which can give information regarding the shape of urine flow, voided volume, flow time, maximum flow rate (Qmax), and average flow rate. Electromyographic (EMG) activity of the urethral sphincter and pelvic floor musculature can also be studied during voiding. The sphincter activity during voiding, which is absent normally, suggests dysfunctional voiding. It must be pointed out, however, that there is a weak correlation between urinary flow and clinical response to therapy, probably due to fact that only a minority of children with urinary incontinence have a failure of the emptying function of the bladder, which is assessed by uroflowmetry. Urine flow patterns generally correlate with the etiology of daytime urinary incontinence. Another parameter that can be obtained during this study is bladder capacity. It can be decreased in children with overactive bladders and increased in those with underactive bladders [8].

Cystometry is performed by simultaneously measuring bladder pressure via a transurethral catheter, intra-abdominal pressure via a transrectal probe, and sphincter activity via perineal patch electrodes. The bladder is filled at a slow rate while pressure is being monitored continuously. Variables including presence of overactive contractions, bladder filling pressures, leak point pressure, voiding pressure, post-void residual volume, and relaxation of the sphincter muscle are recorded. These data allow thorough evaluation of LUT function [5]. This investigation detects abnormalities during the filling as well as voiding phase and can differentiate between overactive bladder, a filling phase abnormality that is more common, and dysfunctional voiding, a consequence of sphincter dysfunction or pelvic floor musculature contraction during voiding. In addition, it can detect voiding abnormalities in most children with daytime incontinence who do not respond to treatment. It is mainly indicated in more challenging cases with a known or suspected neurologic lesion, severe LUTD with evidence of renal injury or hydronephrosis, high imperforate anus, and abnormalities in urinary tract anatomy [8].

## 4. Differentiation of Specific LUTD Subtypes

Specific LUTD subtypes, such as detrusor–sphincter dyssynergia, underactive bladder, and non-neurogenic dysfunctional voiding, require specialized diagnostic approaches. These may include invasive and non-invasive urodynamic measurements tailored to the specific characteristics of each subtype [13]. Table 3 presents urodynamic patterns in normal and abnormal bladder conditions [18].

Children with overactive bladder have involuntary detrusor contractions during the filling phase, called DO (mentioned above), characterized by increased bladder pressure. Dysfunctional voiding occurs in children with a neurologic lesion, where it is called detrusor sphincter dyssynergy, or patients without a neurologic lesion, named non-neurogenic dysfunctional voiding. There is abnormal urethral sphincter contraction during voiding in all of them but normal other urodynamic parameters, as shown in Table 3. Children with an underactive bladder have increased bladder capacity and incomplete bladder emptying during voiding due to decreased detrusor contraction but with normal sphincter function during both the filling and voiding phases [8].

Table 4 presents, in more detail, some of the most common and clinically important functional voiding problems, such as detrusor overactivity, dysfunctional voiding, severe bladder–bowel dysfunction (previously known as the Hinman–Allen syndrome or non-neurogenic neurogenic bladder) and giggle incontinence [5].

## 5. Chronic Kidney Disease and Prognosis

Pediatric LUTD, if left unmanaged, can lead to chronic kidney disease, underscoring the importance of early detection and intervention. Regular monitoring of renal function through urinalysis, serum creatinine levels, and imaging studies is essential for assessing prognosis and tailoring therapeutic strategies [19]. Infants with neurogenic bladder dysfunction (NBD) have a normal upper urinary tract at birth; however, most of them will develop progression of kidney dysfunction and progress to chronic kidney disease (CKD) later in life without proper management because of high intravesical pressures due to poor bladder compliance, DO against a closed sphincter or detrusor sphincter dyssynergia. In order to preserve kidney function, NBD must be treated as soon as possible. Clean intermittent catheterizations (CIC), combined with oral oxybutynin, either orally or intravesically, is the cornerstone of treatment. The new anti-muscarinic drugs have better side effects profile and high tolerability but lack proven efficacy and safety in children. Kidney dialysis or transplantation are the last resort for treatment of NBD in children with advanced CKD when there was no response to other treatments [20].

## 6. Treatment Options in Pediatric LUTD

Children who are distressed by LUTD, have bothersome symptoms, or are at risk for recurrent UTIs, with or without additional risk for kidney damage, should be treated. Patients with abnormal voiding charts, frequent or seldom voiding, with mild urgency or occasional mild urinary incontinence, who are not bothered by these symptoms need no treatment. Table 5 presents basic principles of treatment of most common voiding problems [5,21,22,23].

Basic urotherapy is a cornerstone of treatment of several types of LUTD in children. It includes advice for proper fluid intake and regular voiding at the toilet, approximately every two hours (during daytime), no matter how strong the desire to void is. Children should relax during voiding and support their feet and thighs and take enough time for voiding in order to empty the bladder. It is usually feasible above five years of age and should be followed for at least two months before assessing treatment success [5]. Basic urotherapy has about a 50% success rate in terms of incontinence resolution [20], while in others, inadequate adherence to the voiding schedule, with or without associated constipation treatment, seems to cause treatment failure. In addition, a child psychologist or psychiatrist, in cases of associated neuropsychiatric problems, needs to be included in treatment [5].

## 7. Urinary Incontinence and Quality of Life

Pediatric urinary incontinence, often a consequence of LUTD, profoundly impacts a child’s quality of life (QOL). Beyond physical discomfort, incontinence can lead to social and psychological challenges, affecting a child’s self-esteem and interpersonal relationships. Addressing the psychosocial aspects of LUTD is crucial in providing comprehensive care [3]. Negative impact of urinary incontinence upon children’s QOL was proven in a study on forty children, aged 5–11 years with non-neurogenic daytime wetting, where the association between LUTD, measured by the Dysfunctional Voiding Symptom Score and QOL, measured by the Pediatric Urinary Incontinence QOL tool, was evaluated. In addition, the relationship between parent and patient’s responses were tested and compared [24]. An example of the above-mentioned Dysfunctional Voiding Symptom Score (DVSS) is presented separately, in Appendix A, designated as Appendix A [25].

In another study, children were asked to grade stressful events, and wetting was found to be third behind losing a parent and going blind [26].

## 8. Conclusions

Pediatric LUTD is a multifaceted condition with significant clinical implications. Understanding its association with urinary tract infections, renal function, and its impact on a child’s quality of life is crucial for comprehensive care. The interplay between the lower urinary tract and bowel function, coupled with common conditions such as detrusor overactivity and voiding dysfunction, requires a nuanced diagnostic approach. Utilizing a comprehensive toolbox, including voiding diaries, urodynamics, and various imaging modalities, is essential for accurate evaluation and tailored management. Early detection and intervention are paramount to prevent complications such as chronic kidney disease and ensure a favorable prognosis for affected children.

## Figures and Tables

**Table 1 biomedicines-12-00945-t001:** Definitions of symptoms of bladder dysfunction.

Symptom of Bladder Dysfunction	Definition
Daytime frequency	increased: voiding more than seven times during waking hours; decreased: fewer than four voids
Pollakiuria	abnormally frequent small voids in a previously toilet-trained child but with no polyuria or UTI
Incontinence	uncontrolled leakage of urine (continuous or intermittent)
Urgency	the sudden and unexpected experience of an immediate need to void
Nocturia	awakening to void at night
Hesitancy	difficulty in the initiation of voiding in children who have achieved bladder control regardless of age
Straining	the use of abdominal pressure by the child to initiate and maintain voiding
Weak stream	the observed ejection of urine with a weak force
Intermittent stream	a voiding stream of urine occuring in several discrete bursts rather than the normal continuous stream (a normal physiologic pattern in children less than four years of age)
Dysuria	burning or discomfort during voiding
Holding maneuvers	observed behavior used to either postpone voiding or suppress urgency, such as standing on tiptoe, forcefully crossing the legs, or squatting with a hand pressed into the perineum (Vincent’s curtsy) in children with bladder control regardless of age.
Postmicturition dribbling	involuntary urine leakage immediately after completion of voiding in children with bladder control regardless of age

Abbreviation: UTI—urinary tract infections.

**Table 2 biomedicines-12-00945-t002:** Parts of physical examination [8].

Part of Body Examined	Typical Findings
lower back	cutaneous signs of occult spinal dysraphism or sacral agenesis (presacral dimple, hair patch, lipoma, asymmetric gluteal cleft)
neurologic examination	lower extremity strength and deep tendon reflexes, gait, fine-motor coordination, perineal and anal sensation, rectal tone assessment.
external urological and perianal examination	meatal stenosis in boys, labial adhesions in girls (may cause bladder outlet obstruction); signs of skin excoriation or redness (may indicate continuous or severe urinary leakage)
perianal inspection	position of the anus, the presence of gluteal cleft deviation, dermatitis and perianal fissures, feces or hemorrhoids
abdominal examination	tenderness due to colonic distension secondary to fecal impaction
digital rectal examination	rectal distension (full of stool), information about perianal sensation, tone and function of the anal sphincter
urinary stream observation	signs of LUTD: hesitancy, dribbling, weak urinary stream or intermittency of voiding, especially if observed during voiding

Abbreviation: LUTD—lower urinary tract dysfunction.

**Table 3 biomedicines-12-00945-t003:** Urodynamic patterns in normal and abnormal bladder conditions.

	Filling of the Bladder		Empying of the Bladder	
	Detrusor Contractions	Sphincter Activity	Detrusor Contractions	Sphincter Activity
Normal	−	+	+	−
Abnormalities, discovered during urodynamic investigation				
Overactive bladder	+	+	+	−
detrusor–sphincter dyssynergy	−	+	+	+
Non-neurogenic dysfunctional voiding	−	+	+	+
Underactive bladder	−	+	Insufficient, incomplete emptying	−

Abbreviations: + present; − absent.

**Table 4 biomedicines-12-00945-t004:** Most common and clinically important functional voiding problems [5].

Voiding Problem	Special Characteristics
detrusor overactivity	most common cause of daytime incontinence unknown etiology often associated with a positive family history, constipation, and neuropsychiatric disorders such as ADHD urgency symptoms and incontinence present holding maneuvers are common benign condition, spontaneous resolution common over time if bothered (usually over age of six ys), they should be treated
dysfunctional voiding	may be a learned behavior in children with DO in order to manage the involuntary detrusor contractions or the first sign of previously undetected neurogenic bladder dysfunction uroflow study shows staccato pattern and residual urine may be benign with spontaneous resolution it may lead to recurrent UTIs and hydronephrosis secondary to incomplete bladder emptying and, in most severe cases, to kidney damage
severe BBD	voiding dysfunction + elevated detrusor filling +/− voiding pressures, without spinal cord abnormality may result in chronic urinary retention unknown etiology, most likely heterogeneous unfavorable prognosis without spontaneous resolution often causes recurrent UTIs may even lead to end-stage renal disease if left untreated
giggle incontinence	unknown etiology incontinence only when laughing, usually affects teenage girls LUT function is completely normal Must be differentiated from DO (leakage occurs easily when the child’s focus is lapsing) and stress UI (leakage occurs with pelvic floor laxity during increased intra-abdominal pressure) can be debilitating, without spontaneous resolution

Abbreviations: ADHD—attention deficit hyperactivity disorder; ys—years; DO—detrusor overactivity; UTI—urinary tract infection; BBD—bladder-bowel dysfunction; LUT—lower urinary tract; UI—urinary incontinence.

**Table 5 biomedicines-12-00945-t005:** Basic principles of treatment of most common forms of voiding problems [5,21,22,23].

Voiding Problem	Basic Principles of Treatment
Overactive bladder	treatment of associated constipation, if present basic urotherapy + bladder relaxants or anticholinergics (not if incomplete bladder emptying). Only oxybutynin approved for use in children, alternative drugs seem to have better side-effect profiles, but more research is needed before their implementation in pediatric use side effects of anticholinergic therapy: constipation (most common), central nervous system effects (mood swings, aggression), decreased saliva production, incomplete bladder emptying with increased risks for UTIs the effect of therapy is assessed after at least one month TCES, ISNS, and intradetrusor botulinum toxin injections may be used as third-line therapies in cases of severe treatment resistance
dysfunctional voiding	Treatment indicated in presence of UTIs or kidney damage neurogenic bladder or severe BBD need to be excluded urotherapy with biofeedback is first-line therapy—the goal is to teach a child how to distinguish between a coordinated voiding and associated sphincter contraction in cases of treatment failure, α-blocker drugs can be used; however, CIC (as in patients with neurogenic bladders), should be initiated, especially in children at risk for UTIs, kidney damage, or severe incontinence
severe BBD	CIC is foundation of treatment, with the purpose of protection of kidney function, UTI prevention, and gain of social continence in cases of treatment failure, α-blocker drugs, intravesical anticholinergics, intradetrusor botulinum toxin, or, in most severe cases, surgical bladder augmentation and reconstruction of the bladder outlet are indicated
giggle incontinence	difficult to treat, sufficient expertise needed advanced urotherapy (biofeedback) or off-label use of central nervous system stimulant drug (methylphenidate, for example) seem to provide the highest success rate

Abbreviations: TCES—transcutaneous electrical stimulation; ISNS—invasive sacral nerve stimulation; CIC—clean intermittent catheterization; UTI—urinary tract infection; BBD—bladder-bowel dysfunction.

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
