# Peer review of "Pediatric Lower Urinary Tract Dysfunction: A Comprehensive Exploration of Clinical Implications and Diagnostic Strategies"

_biomedicines, 2024, doi:10.3390/biomedicines12050945_

Round 1
Reviewer 1 Report
Comments and Suggestions for Authors
In the manuscript, entitled »Pediatric Lower Urinary Tract Dysfunction: A Comprehensive Exploration of Clinical Implications and Diagnostic Strategies« submitted to Biomedicines for a potential publication, the author presents a review article investigating the lower urinary tract dysfunction in children and its impact on their health. I am of opinion that in the present form, the article is not good enough to be published and should be substantially improved before reevaluation. My comments are presented below.
The comments:
1. The whole spectrum of important functional voiding problems has to be presented in more detail, from both clinical and diagnostilcal point of view.
2. Their comparison would be of value, preferably in the form of Tables.
3. The type of a review has to be cited.
4. A short presentation of treatment options should be added, including the new ones.
5. Quality of life of children with lower urinary tract dysfunction (LUTD) should be presented in more detail.
6. The interplay of bowel function and urinary tract should be clearly shown.
7. Some structural questionnaires for voiding problems have to be presented as well as their usefulness.
8. How a child abuse in the context of LUDS is suspected? And what has to be done to exclude it?
9. The history, clinical examination and observations in infant with suspected LUTD should be delineated in more detail.
10. Is there any place for voiding sonography and CEUS (contrast enhanced ultrasound) in the diagnostic evaluation? Comment!
11. The importance of chronic kidney disease in LUTD has to be further discussed.
12. There are a lot of very old references cited and the review is based mostly on UpToDate novelties, therefore a new knowledge has to be add with newer references and a wider view.
Comments on the Quality of English Language
Minor editing of English language required.
Reviewer 2 Report
Comments and Suggestions for Authors
The article provides a comprehensive overview of pediatric LUTD, covering its clinical relevance, diagnostic approaches, and implications for prognosis. It offers valuable insights into the complex interplay between lower urinary tract dysfunction and various clinical outcomes. The thorough discussion of diagnostic modalities and differentiation of specific LUTD subtypes enhances the practical utility of the article for healthcare professionals involved in the management of pediatric patients with LUTD. My only suggestion would be to add the clear aim of the article to the introduction section and to describe in more details literature search strategy.
Reviewer 3 Report
Comments and Suggestions for Authors
This contribution reviewed various aspects of pediatric lower urinary tract dysfunctions, ranging from the association of urinary tract infections, renal function variations, association with bowel functions, likely pathogenesis, and diagnosis workup approach. This contribution provided a detailed approach that could help the clinician to diagnose and subsequent clinical management for these patients, leading to a favorable outcome for the affected cases.
Author Response
Response to Reviewer 3 Comments:
Dear Madam/Sir,
Thank you for your comments:
This contribution reviewed various aspects of pediatric lower urinary tract dysfunctions, ranging from the association of urinary tract infections, renal function variations, association with bowel functions, likely pathogenesis, and diagnosis workup approach. This contribution provided a detailed approach that could help the clinician to diagnose and subsequent clinical management for these patients, leading to a favorable outcome for the affected cases.
Response: thank you, I appreciate your positive and constructive approach to this article.